# The Sweet Relationship between the Endometrium and Protein Glycosylation

**DOI:** 10.3390/biom14070770

**Published:** 2024-06-27

**Authors:** Linyu Zhang, Ying Feng, Yue Zhang, Xinrui Sun, Qianhong Ma, Fang Ma

**Affiliations:** 1Center for Translational Medicine, Key Laboratory of Birth Defects and Related Diseases of Women and Children (Sichuan University), Ministry of Education, West China Second University Hospital, Sichuan University, Chengdu 610041, China; 2Department of Obstetrics and Gynecology, West China Second Hospital, Sichuan University, Chengdu 610041, China; 3West China School of Basic Medical Sciences & Forensic Medicine, Sichuan University, Chengdu 610041, China; 4Key Laboratory of Birth Defects and Related Diseases of Women and Children (Sichuan University), Ministry of Education, West China Second University Hospital, Sichuan University, Chengdu 610041, China

**Keywords:** glycosylation, endometrial receptivity, endometriosis, endometrial cancer

## Abstract

The endometrium is an important part of women’s bodies for menstruation and pregnancy. Various proteins are widely expressed on the surface of endometrial cells, and glycosylation is an important post-translational modification of proteins. Glycosylation modification is closely related not only to endometrial receptivity but also to common diseases related to endometrial receptivity. Glycosylation can improve endometrial receptivity, promote embryo localization and trophoblast cell adhesion and invasion, and contribute to successful implantation. Two diseases related to endometrial receptivity include endometriosis and endometrial cancer. As a common benign disease in women, endometriosis is often accompanied by an increased menstrual volume, prolonged menstrual periods, progressive and aggravated dysmenorrhea, and may be accompanied by infertility. Protein glycosylation modification of the endometrial surface indicates the severity of the disease and may be an important pathogenesis of endometriosis. In cancer, glycosylation modifications on the surface of tumor cells can be a marker to distinguish the type and severity of endometrial cancer. This review highlights the role of protein glycosylation in embryo–maternal endometrial dialogue and explores its potential mechanisms in diseases related to endometrial receptivity, which could provide a new clinical approach for their diagnosis and treatment.

## 1. Introduction

Various proteins are widely expressed on the surface of endometrial cells, and glycosylation is an important post-translational modification of proteins. Glycosylation refers to the transfer of glycan to a specific residue on a protein by glycosyltransferase to form a glycosidic bond, which mainly occurs in the endoplasmic reticulum and Golgi apparatus [1]. N-linked glycosylation and O-linked glycosylation are the two main types, and fucose and sialic acid (Sia), as well as other types of sugars, can be added to N-glycan and O-GalNAc glycan. Because fucose and Sia, two terminal modifications, have been extensively studied in endometrial-related diseases, they are also discussed in separate sections. 

Endometrial receptivity is closely related to the development of many diseases. Clinical data estimates indicate that one-third of implantation failures are due to the embryo itself, and another two-thirds are due to poor endometrial respectability and inconsistent dialogue between the embryo and the endometrium [2,3,4]. Glycosylation is a significant factor affecting endometrial receptivity. Therefore, understanding how glycosylation affects endometrial receptivity is important for preventing miscarriage. Additionally, endometrial receptivity-related diseases also include endometriosis, fibroids, endometritis, endometrial cancer, choriocarcinoma, and endometrial hyperplasia [5,6,7,8,9,10]. The occurrence and development of all these diseases are related to dysregulated glycosylation [5,6,7,8,9,10]. Endometriosis is one of the most common gynecological diseases affecting reproductive functions, such as ovarian function, oocyte quality, and embryo development and implantation in women [11]. It affects 6–10% of women of reproductive age and causes infertility in up to 30% of women [12]. In recent years, a growing number of studies have focused on changes in glycosylation in endometriosis. Among malignant tumors of the female reproductive tract, endometrial cancer is the most common, and glycosylation is inextricably associated with tumor cells. Glycosylation has a wide influence on the proliferation, adhesion, migration, and invasion of endometrial cancer cells and is also one of the indicators of the degree of malignancy [13,14,15,16,17]. Hence, summarizing the relationship between glycosylation and endometrial receptivity-related diseases has important clinical value for the diagnosis and treatment of these diseases.

This review focuses on the relationship between N-linked glycosylation, O-linked glycosylation, fucose and Sia modification, and endometrium receptivity-induced miscarriage, endometriosis, and endometrial cancer.

## 2. A Brief Introduction to Protein Glycosylation

Proteins are mainly modified by N-linked glycosylation and O-linked glycosylation, and fucose and Sia are often used as the branch chain ends of N-glycan and O-glycan; see Figure 1.

N-glycan is covalently attached to proteins on asparagine (Asn) residues via N-glycosidic bonds, which is called the N-linked glycosylation modification of proteins. Most membranes and secreted proteins undergo N-glycan modification as they enter the endoplasmic reticulum lumen. Therefore, the loss of N-glycan in proteins may affect protein folding and stability, intercellular recognition, and signaling [18]. N-glycan is synthesized through the endoplasmic reticulum and Golgi apparatus and finally reaches the endosome or plasma membrane surface of the cell [19]. The early stage is mainly regulated by the α-glucosidase I (MOGS) of the endoplasmic reticulum and the α-mannosidase (MANEA) of the Golgi apparatus, and the later stage is mainly regulated by the N-acetylglucosamine transferase GLNAC-Ti (MGAT1). The protein N-glycans of female endometrial epithelial cells and stromal cells play a key role in the normal function of endometrial implantation and the maintenance of pregnancy [20,21].

O-linked glycosylation of endometrial proteins includes O-GalNAc glycan and O-GlcNAc modification. O-GalNAc glycan is attached to the hydroxyl group of serine (Ser) or threonine (Thr) residues of membrane proteins and secreted proteins [22], and O-GlcNAc modification is a dynamic modification of the Ser or Thr hydroxyl group on the nucleus, mitochondria, and cytoplasmic proteins. The synthesis of O-GalNAc glycan is affected by a variety of glycosyltransferases. Only two enzymes regulate O-GlcNAc synthesis: O-GlcNAc transferase (OGT) and β-n-acetylaminoglycosidase (OGA) [23]. The ends of O-GalNAc glycan may contain alpha bonds of fucose and Sia, which are recognized by lectins. Some sugar residues or their modifications may mask potential antigens or receptors [24].

Fucose is a 6-site deoxygenated and L-shaped 6-carbon sugar used to form various glycoproteins in mammalian cells [22]. Fucosyltransferase (FUT) is an enzyme that catalyzes the transfer of guanosine diphosphate (GDP) fucosylated to modify target proteins [25]. In approximately 90% of mammals, GDP fucose is produced by the neonate pathway [26]. It is catalyzed by L-fucose kinase and GDP-L-fucose pyrophosphorylase to convert fucose to GDP-fucose through a two-step mechanism. Once synthesized, it is transported to the Golgi cavity or endoplasmic reticulum for use by the FUT [27]. FUT subsequently catalyzes a shift of fucose from donor molecules to oligosaccharides, glycoproteins, and glycolipids in eukaryotes and prokaryotes [28]. Terminal fucosylation is a common terminal modification in many N-glycan and O-GalNAc glycans. Ten FUTs (FUT1–7 and FUT9–11) are known to add terminal fucose to these oligosaccharide chains. These FUTs are localized to the Golgi apparatus. Lewis antigens are produced by adding fucose to the terminal galactose and subterminal GlcNAc of oligosaccharides [22]. Protein-associated lesions are involved in many biological processes during development, such as cell adhesion, inflammation, white blood cell transport, and fertilization [25].

In 1936, Gunnar Blix first isolated a substance from salivary mucin and named it Sia after the Greek word for saliva [24]. Sia is a negatively charged nine-carbon sugar with a unique structure. Sialtransferase is a glycosyltransferase that catalyzes the transfer of Sia from the donor substrate CMP-Sia [29] to a non-reducing position at the end of oligosaccharide chains, producing α-2, 3-chain ST3GAL, α-2, 6-chain ST6GAL, or α-2, 8-chain ST8Sias [30]. Sialidase is encoded by the NEU gene, of which there are four in the human genome, including NEU1, NEU2, NEU3, and NEU4. The Sia synthesized by humans is N-acetylneuraminic acid (Neu5Ac). The sum of the various sialoglycans in a cell is called its “Sia.” As mentioned earlier, Sia is usually the most terminal sugar of N-glycan and O-glycan. Sia can regulate the biophysical environment, mask potential glycans, and mediate the specific recognition of complementary Sia-binding proteins in biological processes [24].

## 3. Glycosylation and Endometrial Receptivity

The study by Carson, D. D. (2002) showed that glycosylation can improve endometrial receptivity [31] (Figure 2). Endometrial receptivity refers to the ability of the endometrium to promote embryo attachment, implantation, and subsequent growth [32]. Acceptance of the endometrium is an important prerequisite for embryo implantation, which is a key step in pregnancy.

### 3.1. N-Linked Glycosylation

Abundant N-linked glycosylation-modified proteins are distributed on the surface of endometrial cells and regulate their proliferation and apoptosis. LIFR is a 197 kDa glycoprotein with 18 potential N-glycan junction sites [21]. It has been reported that the LIF/LIFR axis regulates endometrial and embryonic function during human reproduction [33]. LIF acts on endometrial epithelial cells by binding to LIFR heterodimers composed of gp130 and LIFR transmembrane proteins. The STAT3, mitogen-activated protein kinase, and protein kinase C pathways are then activated to promote blastocyst adherence to the endometrium [34]. LIFR and gp130 mRNA expressed in human blastocysts interact with endometrial epithelium [35], and LIF promotes endometrial stromal cell exocytosis [36]. Importantly, in infertile women, the levels of LIF, LIFR, and gp130 in the endometrium were significantly reduced [37], suggesting that LIF and its receptors influence female pregnancy. LIF was highly expressed in the cervical epithelium of mice and humans at the receiving stage. Therefore, the LIF expression in the cervical epithelium or serum (as measured using an enzyme-linked immunoassay) can be used as a less invasive marker for predicting endometrial receptivity compared to scraping the endometrial tissue [38], thereby avoiding potential endometrial injury [39]. ENPP3 is a protein that is active against extracellular nucleotides, ATPase, and ATP-hydrolyzing pyrophosphatase [40]. Boggavarapu et al. (2016) found that after de-N-linked glycosylation, the molecular weight of ENPP3 decreased from 165 kDa to only 110 kDa, indicating that ENPP3 underwent N-linked glycosylation modification [41]. ENPP3 peaks in the middle stage of endometrial secretion, and the overexpression of ENPP3 can upregulate LIF and integrin-β3 in endometrial cells, which is conducive to embryo adhesion and invasion [42]. 

### 3.2. O-Linked Glycosylation

The functions of O-GalNAc glycans are varied depending on their structure and density, as well as the proteins to which they are attached. Mucin (MUC) is the glycoprotein that carries the largest amount of O-GalNAc glycan. Mucins are highly glycosylated O-GalNAc-modified sialoglycoproteins, and their dense anionic charge and hydrophilicity make them effective as moisturizing and protective barriers on tissue surfaces in contact with the environment. Mucin 1 (MUC1), a member of the mucin family, is a product of the highly glycosylated, hormone-regulated glandular and luminal epithelium of the endometrium, with adhesion and anti-adhesion properties [43]. During the secretion of MUC1, sialylation of MUC1 expression by endometrial cells increases the formation of the receptive endometrium to prepare for implantation. In the middle stage of implantation, MUC1 will be secreted into the glandular cavity and co-exist in the uterine fluid [44]. This suggests that the detection of MUC1 in female uterine fluid may indicate the implantation of an embryo. On the other hand, the cell adhesion properties of MUC1 may contribute to promoting the binding of trophoblast cells to the endometrial cell surface through the L-selectin/sialyl Lewis X (sLeX) adhesion system after implantation [45]. MUC1 is a large, highly glycosylated, hormone-regulated product of the endometrial glandular and luminal epithelium with cell surface-associated and secreted isomers. From the proliferative phase to the early secretory phase, the abundance of MUC1 mRNA coding increased by about six times. In women with recurrent spontaneous miscarriage, the mid-secretory levels of the MUC1 core protein and muco-associated glycans are reduced. Reduced epithelial secretory function and resulting changes in the composition of uterine fluid are characteristic of the endometrium in patients with recurrent miscarriage [46]. O-GlcNAc modification is associated with early pregnancy processes, such as trophoblast differentiation [47], embryo implantation [48], and endometrial receptivity. Han et al. (2019) found that an increase in O-GlcNAc affected endometrial receptivity by enhancing the proliferation, adhesion, migration, and invasion of endometrial cells [49]. A recent study showed that O-GlcNAc levels were higher during endometrial secretion than during the proliferative and menstrual periods [49]. During the secretion phase, the expression of glucose transporter 1 (GLUT1) is elevated [50]. GLUT1 can increase glucose uptake by endometrial cells into the hexosamine biosynthesis pathway (HBP), and the activation of HBP increases O-GlcNAcylation [51,52]. It is speculated that O-GlcNAc can bypass the pentose phosphate pathway and HBP through reprogrammed metabolism at the implantation window. However, O-GlcNAc can compensate for glycolysis through GLUT1-mediated glucose uptake and intracellular glycerol transport mediated by aquaporin 3, thereby maintaining physiological requirements and regulating endometrial receptivity [52]. In addition, OGA inhibitors can inhibit the hydrolysis of O-GlcNAc, thereby increasing the level of O-GlcNAc [53].

### 3.3. Fucosylation

The glycobiology of implantation suggests that fucosylation is a functional event at the maternal–fetal interface [31]. FUT4 is a key enzyme in α-1, 3-fucose-mediated glycoprotein biosynthesis and is specifically expressed in different stages of the mammalian endometrium [54]. Compared to the human proliferative stage, more FUT4 is expressed in the secretory stage. This facilitates the establishment of endometrial receptivity and is, therefore, considered a marker of it [55,56]. In addition, FUT4 is a leukocyte-associated enzyme involved in the synthesis of selectin ligands, such as sLeX and Lewis Y (LeY) [55]. LeY is a bis fucose oligosaccharide [57] whose α-1, 3-fucose is partially catalyzed by its key synthetase FUT4 [58]. Therefore, FUT4 can regulate embryonic adhesion by regulating the synthesis of LeY on the endometrial epithelial surface, and its overexpression can promote the recognition and adhesion of endometrial cells to embryonic cells [59]. Integrin αvβ3 carries LeY oligosaccharides in human epithelial cells (RL95-2), and LeY can bind to glycoproteins in the plasma membrane of endometrial cells, affecting the establishment of endometrial receptivity [60,61]. In addition, LeY expression is low in patients with a thin endometrium and repeated implantation failure, suggesting that LeY may affect pregnancy by affecting the physiological state of the endometrium [62]. The serum levels of miR-200 family members in infertility and miscarriage patients were significantly higher than those in healthy non-pregnant and early pregnant women. miR-200c is the most sensitive diagnostic criterion for infertility. Women with infertility and miscarriage had lower serum FUT4 levels than healthy non-pregnant and early-pregnancy women. miR-200c targets and inhibits the expression of FUT4, leading to uterine receptivity dysfunction. miR-200c reduced the α1.3-fucosylation modification on the glycoprotein CD44 and further inactivated the Wnt/β-catenin signaling pathway. miR-200c blocks the formation of uterine receptivity by targeting α1.3-fucoglycation modifications that reduce FUT4 and CD44 [63]. Together, miR-200c and FUT4 may serve as potential markers of endometrial receptivity as well as useful diagnostic and therapeutic targets for infertility. Protein O-fucosyltransferase 1 (poFUT1) is a key enzyme of the protein O-fucose. The poFUT1 levels in stromal cells were higher in the secretory phase of the menstrual cycle than in the proliferative phase, whereas the poFUT1 levels were lower in the secretory stromal cells of miscarriage patients than in the proliferative phase. poFUT1 increased the O-fucosylation modification of Notch1 in human embryonic stem cells (hESCs) and activated the Notch1 signaling pathway. Activated Notch1, as a specific trans-factor of PRL and IGFBP1 promoters, can enhance the transcriptional activity of PRL and IGFBP1, thereby inducing the decidualization of hESCs [64]. The relationship between FUT and pregnancy-associated plasma protein-A (PAPPA) also affects endometrial receptivity [65]. PAPPA is a high-molecular-weight zinc-bound metalloproteinase [66]. Yu et al. (2017) reported that PAPPA from maternal serum samples, recombinant human PAPPA, or trophoblastic-conditioned media can promote endometrial receptivity in vitro. Mechanistic studies have shown that recombinant human PAPPA may upregulate the expressions of FUT1 and FUT4 in human endometrial cells through the insulin-like growth factor 1 receptor/phosphocreatine 3 kinase/AKT signaling pathway, thereby increasing the levels of α-1, 2-fucose, α-1, 3-fucose, and α-1, 6-fucose [65]. In summary, PAPPA promotes endometrial receptivity by increasing fucose levels, suggesting that fucosylation plays an important role in successful pregnancy. The treatment of mice with antibodies blocking PAPPA inhibited embryo adhesion in vivo and the diffusion of implanted mouse embryos in vitro and further reduced the level of N-fucose in the endometrium of pregnant mice [65]. Decreases in the PAPPA level further affect the fucosylation level of the endometrium, thus affecting the adhesion and invasion of trophoblast cells, suggesting that protein glycosylation has a significant effect on embryo implantation [65]. This evidence is still being studied in animals and needs to be confirmed in clinical trials.

### 3.4. Sialylation

Recent studies have shown that the effect of sialylation on endometrial receptivity mainly involves the interaction between sLeX and selectin [66,67,68]. ST3GaL-3, 4,6 glycosyltransferases act on the N-acetyllactosamine substrate type II to add terminal Sia to form sialyllactosamine. Alpha (1,3) FUT acts on sialylactosamine to form the sLeX antigen. sLeX is the main ligand of selectin (L-selectin, E-selectin, and P-selectin) [69] and is involved in fertilization [70] and implantation [66]. In maternal–fetal dialogue, L-selectin expressed by trophoblast cells and sLeX expressed by endometrial cells recognize each other, promoting embryo implantation, which is a major marker of endometrial receptivity [68]. Zhang et al. (2009) found that the overexpression of FUT7 can increase the biosynthesis of sLeX oligosaccharides, which is conducive to the establishment of endometrial tolerance in vitro and in vivo [71]. Terminal sialylation is the most widely studied in the field of cancer. It can be used as a biomarker of cancer and may alter the phenotype of cancer cells [24] and the biological behavior of tumors. In the field of reproduction, Sia terminal modification is widely present at the maternal–fetal interface, so whether sialylation can improve endometrial receptivity by increasing embryo or endometrial quality is worthy of further study. The Sialyl 6-sulfo Lewis X determinant (also known as MECA-79 epitopes) is another ligand of L-selectin that mediates cell–cell interactions [72]. The expression of MECA-79 on the apical surface of thin endometrial luminal epithelial cells was 1.3 times that of normal endometrial cells. This difference may adversely affect the establishment of endometrial receptivity [62]. In addition, Ziganshina et al. (2021) suggested that reduced endometrial acceptance in infertile women was related to the decreased expression of MECA-79 in the calyx of thin endometrial epithelial cells [62]. Therefore, MECA-79 is considered to be one of the most relevant markers of endometrial receptivity [73]. One of the important biological functions of sLeX is its role in leukocyte adhesion and exosmosis. IL-1β promotes the expression of sLeX on the cell surface in endometrial RL95-2 cells by increasing the level of transcription (FUT3). The fetomaternal interfacial inflammatory microenvironment can regulate the glycosylation pattern of endometrial cells during implantation. Therefore, IL-1β can significantly induce sLeX by increasing FUT3 expression and promoting trophoblast adhesion during implantation [67].

The GEO database (GSE113790) for recurrent miscarriage and induced miscarriage has documented 15 differential genes related to glycosylation, and existing studies have shown that CD69 and OLR1 are related to RSA (Figure 2).

CD69 is involved in lymphocyte proliferation and functions as a signal-transmitting receptor in lymphocytes, natural killer (NK) cells, and platelets. Recurrent spontaneous miscarriage (RSA) and unexplained in vitro fertilization (IVF) failures are controversial issues that may be related to the immune system. The percentage of NK cells in RSA patients and women with IVF failure increased significantly compared to healthy multiple births and successful IVF controls. Furthermore, the overall expression of CD69 in NK cells increased significantly in both groups compared to the control group. Therefore, increased NK cell CD69 expression can be considered an immunological risk marker for RSA and IVF failure. Unexplained recurrent miscarriage was associated with significantly reduced expression of T cell co-receptor CD8 and tissue-resident marker CD69 [74]. 

Oxidized low-density lipoprotein receptor 1 (OLR1) mediates the recognition, internalization, and degradation of oxidatively modified low-density lipoprotein (ox-LDL) through vascular endothelial cells. The expression of OLR1 in pregnant patients with unexplained recurrent miscarriage was higher than that in non-pregnant uRM patients. This study shows that the OLR1 expression levels in the peripheral blood of uRM women are correlated, suggesting that these women may have uncontrolled oxidative stress during their first trimester of pregnancy [75].

## 4. The Relationship between Endometriosis and Glycosylation

Endometriosis is commonly thought to be caused by retrograde menstruation and is characterized by the presence of endometrium tissue (including epithelial, stromal, and muscle cells) outside the uterus [76]. Women with endometriosis often have increased menstrual volume, prolonged periods, and progressive dysmenorrhea, which may be accompanied by infertility [77]. Early detection and diagnosis of endometriosis can help in early treatment and reduce the symptoms of patients [78].

### 4.1. N-Linked Glycosylation

N-glycan in the serum may indicate endometriosis. A study reported that the N-linked glycosylation levels of the serum total glycoprotein and IgG in patients with endometriosis were different from those in the normal endometrial group. In the serum, N-glycopeptide GP24 (double-antenna dichotomized monosialoglycan) was significantly increased in the samples from patients with endometriosis [79]. Additionally, compared to healthy women, IgG isolated from the advanced endometriosis group showed a significantly higher expression of highly branched multi-antennary N-glycans. Analysis of the expression of high-branched chain N-glycan in isolated serum IgG based on lectins may help distinguish between women with advanced endometriosis and healthy women [80]. More importantly, N-glycan on the surface of cells contributes to the formation of endometriosis. Tunicamycin inhibited the N-linked glycosylation of CD44, thus inhibiting the adhesion of endometrial cells to peritoneal mesothelial cells, suggesting a role in the establishment of early endometriosis lesions [81] (Figure 3). 

### 4.2. O-Linked Glycosylation

O-linked glycosylation is also associated with endometriosis. B-GalNAc inhibits the O-linked glycosylation of CD44, thus significantly inhibiting the adhesion of endometrial cells to peritoneal mesothelial cells, suggesting that the O-linked glycosylation of CD44 plays a role in the establishment of early endometriosis lesions [81]. The O-linked glycosylation and high-branched N-glycan expression of IgG based on lectin-isolated serum may help distinguish women with advanced endometriosis from healthy women [80] (Figure 3).

### 4.3. Fucosylation

Dolichos biflorus lectin (DBA) is a specific GalNAc. Compared to the control group, DBA-endometrial binding was significantly reduced in patients with endometriosis. The secretory glycosylation of women with advanced endometriosis was different from that of healthy women, and the fucosylated GalNAc bound to DBA was reduced [8]. The reactivity of serum IgG with the fucose-specific lectins Aleuria aurantia lectin (AAL), Lens culinaris agglutinin (LCA), and Lotus tetragonolobus agglutinin (LTA) can be used as an effective diagnostic and clinical tool to distinguish advanced endometriosis [82] (Figure 3).

### 4.4. Sialylation

Endometriosis is an inflammatory disease that is difficult and invasive to diagnose and, therefore, requires non-invasive diagnostic methods and parameters [83]. The sialylation and galactosylation of serum IgG were determined using MAA (α2,3-Sia) and SNA (α2,6-Sia) lectins. ROC and cluster analysis results showed that serum IgG MAA reactivity, sialylation, and galactosylation factors could be used as supplementary parameters for the diagnosis of endometriosis [84] (Figure 3). 

Transcriptome analysis of the GEO database (GSE232713) of endometriosis and normal endometrium revealed 24 differential genes related to glycosylation (Figure 3). Existing studies have indicated that *ACE*, *HPSE*, *HYAL2*, *LMAN2*, and *SULF2* are related to endometriosis. The angiotensin-converting enzyme (ACE) belongs to the peptidase M2 family. Tanshinone type IIA in rats with endometriosis-correlation pain inhibition may be achieved by reducing the expression of E2, ANGII, and AT2. Thus, the dorsal root ganglion (DRG) is blocked, and the threshold of hyperalgesia is raised [85]. The polymorphism of the ACE c.2350A > G and ACE c.240A > T genes may be associated with the development of endometriosis [86].

Heparanase (HPSE) belongs to the endoglycosidase family, which cleaves heparan sulfate proteoglycans (HSPGs) into heparan sulfate side chains and core proteoglycans. Both heparinase (Hpa) and angiopoietin 2 (Ang-2) are expressed in endometriosis. The high expressions of Hpa and Ang-2 in the ectopic endometrium may play an important role in the pathogenesis and development of endometriosis [87]. Heparinase-1 (an endo-glycosidase that degrades heparin sulfate proteoglycans) is expressed in the endometrial tissue of patients with endometriosis. There was no significant difference in the expression of heparanase-1 between the ectopic endometrium and the normal endometrium in women with endometriosis [88].

Hyaluronidase-2 (HYAL2) is a member of the glycohydrolase 56 family. There were no significant differences in the HYAL2 mRNA or protein expression in endometrial epithelial cells (EECs) or stromal cells (ESCs) with and without endometriosis [89], nor in the 4-methylumbelliferone (4-MU)–hyaluronic acid synthesis inhibitors, the expression of the hyaluronic acid (HA) system, or the inhibition of hyaluronidase by endometrial epithelial cells (EECs) or stromal cells (ESCs) on peritoneal mesoepithelial cells (PMCs). 4-MU decreased endometrial cell adhesion, migration, and invasion of PMC and, therefore, is a potential treatment for endometriosis [90]. 

LMAN2 is a vesicular integral membrane protein, VIP36, which plays a role as an intracellular lectin in the early secretory pathway. It interacts with N-acetyl-D-galactosamine and high-mannose-type glycans and may also bind to O-linked glycans. It is involved in the transport and sorting of glycoproteins carrying high-mannose-type glycans. There was no difference in the serum mannose-type glycans of patients with and without endometriosis. Mannose-binding lectins may be involved in the regulation of inflammatory responses, but they do not seem to be involved in the pathogenesis of endometriosis [91].

Extracellular sulfatase 2 (SULF2) exhibits arylsulfatase activity and highly specific endoglucosamine-6-sulfatase activity. The steroid sulfatase (STS) is involved in the synthesis of estradiol (E2), which is the most potent estrogen in the body. The high level of STS activity detected in ectopic endometrium and its correlation with disease severity suggest that STS inhibitors could be used to treat endometriosis [92]. STS was localized in the cytoplasm of cumulus cells, and STS mRNA expression was observed. The STS mRNA expression levels in patients with endometriosis were significantly higher than those in patients without endometriosis. STS may be related to egg quality in patients with endometriosis [93].

## 5. Glycosylation of Endometrial Cancer Cells

Endometrial cancer is a common disease of the endometrium that may be related to reproductive endocrine disorders, estrogen overstimulation, early menarche, late menopause, and infertility. Most women with endometrial cancer are estrogen-dependent and of younger age, have vaginal bleeding and menstrual disorders, and have a history of infertility [94,95,96,97,98]. Some studies have shown that cell-surface N-glycan can be used as a marker to distinguish endometrial cancer types and severity [16,99]. The incidence is increasing every year, including among young people. However, the mechanism of its proliferation and progression is not fully understood. The post-translational modification of proteins is associated with tumorigenesis, and glycosylation is one of the most important types (Figure 4).

### 5.1. N-Linked Glycosylation

Cell-surface glycans play a vital role in biological processes, and changes in their morphology can lead to carcinogenesis. The detection of a complex N-glycan in the core fucose can distinguish primary tumors from non-lymph node metastasis (LNM). The abundance of oligomannan in tumors was higher than that in normal areas, while the abundance of complex N-glycan was lower. In primary tumors with LNM, the abundance of complex core fucosylated N-glycan was reduced [9]. Compared to the control group, the serum levels of N-glucuronic acid were significantly different in patients with endometrial cancer. Total hybrid N-glycans significantly correlated with endometrial cancer differentiation types could effectively classify endometrial cancer into well-differentiated or poorly differentiated subgroups, which supports serum N-glucose-coupled markers as potential markers for endometrial cancer diagnosis and phenotype [100]. The overexpression of N-acetylgalactosaminotransferase-6 (GalNAc-T6) in endometrial cancer cells is associated with a lower invasive profile. Therefore, GalNAc-T6 may be a potential indicator of good prognosis and non-aggressive tumors in patients with endometrial cancer [15,16] (Figure 4).

### 5.2. O-Linked Glycosylation

Abnormal O-linked glycosylation is frequently observed on the surface of tumor cells and is associated with poor outcomes and a poor prognosis in cancer patients. HMMC1 is a human monoclonal antibody that reacts with concentrated and extended core 1-O-glycans as epitopes. HMMC1 may block the cell cycle of phase G (1) by binding to O-glycans on CD166, thereby improving the ability of HMMC1 to target cancer-associated forms of CD166. The invasive growth of HMMC-1 epitope-positive endometrial cancer cells was inhibited [101]. The adhesion ability of endometrial cells to type I collagen was enhanced after treatment with the O-linked glycosylation inhibitor, indicating that glycosylation of the MUC1 extracellular domain can regulate the adhesion properties of cells [102]. A single GalNAc residue attached to Ser or Thr forms the Tn antigen, with levels elevated in cancer cell mucins. The antigen overexpression of Tn and sialyl-Tn (s-Tn) was significantly correlated with cyclooxidase-2 overexpression. There was a significant correlation between s-Tn overexpression and low CD8 infiltration. The high expression of the s-Tn antigen was significantly associated with a poor prognosis [17] (Figure 4).

### 5.3. Fucosylation

According to the monoclonal antibody (MSN-1) against the endometrial cancer cell SNG-II, its recognition antigen is mainly the Lewis (b) antigen, which reacts strongly with MSN-1 (SNG-S group) and weakly with MSN-1 (SNG-W group). The expression of the Lewis (b) antigen in SNG-S was stronger than that in SNG-W, while the expression of a 1-->4-fucosyltransferase (FUT) in SNG-S was higher than that in SNG-W. The activity of 4-FUT is higher than that of SNG-W, so the expression of endometrial cancer-specific FUT may be mainly controlled by α-FUT activity [103]. FUT8 is the only FUT responsible for α 1, 6-linkage fucosing (core-fucosing) and is involved in a variety of physiological and pathophysiological processes, including cancer biology. Compared to normal endometrium, the expression of the FUT8 gene was significantly increased in endometrioid carcinoma. Lowering FUT8 significantly inhibited the proliferation of Ishikawa cells, suggesting that FUT8-induced core lesions may be involved in the proliferation of endometrioid cancer cells [13]. UEA I was combined with L-fucose. We found that UEA I reactive glycoproteins are abundant in endometrial cancer and virtually absent in normal endometrial tissue, even when glycoproteins are strongly expressed. Therefore, UEA I reactive glycoprotein can be a marker of endometrial cancer [99] (Figure 4).

### 5.4. Sialylation

Neu5Ac, sometimes called “NANA,” is the most common Sia in humans. The level of total Sia in patients with endometrial cancer before surgery was significantly higher than that in the control group, while the level of serum Neu5Ac after surgery was significantly lower, suggesting that the serum Neu5Ac level can be used as a tumor marker for the suitability of surgical treatment for early endometrial cancer [104] (Figure 4).

## 6. Conclusions

By reviewing the literature, we found that there are abundant proteins distributed on the surface of the endometrium, and the glycosylation modification of proteins affects the biological behavior of the endometrium. Compared to a large number of studies focusing on the role of proteins in endometrial receptivity, there are relatively few studies on the role of endometrial cell surface glycosylation and intracellular proteins. This may be due to the complexity of the structure of sugar chains [1], the resulting lack of methods for accurately characterizing and quantifying polysaccharides [105], and the high diversity of glycosylated forms in organisms, leading to limitations in detecting glycosylated modifications. Therefore, as an important glycosylation modification of proteins, the glycopeptide structure analysis method and the application of glycoprotein are still worthy of further exploration.

In conclusion, the current research on the correlation between glycosylation and endometrial receptivity provides valuable insights for the clinical diagnosis and management of spontaneous miscarriage. Furthermore, anomalies in glycosylation targets could form the basis for the development of innovative contraceptive approaches. Glycosylation intervention holds promise for the amelioration of ectopic colonization of endometrial cells and the mitigation of endometrial cancer progression. Notably, glycosylation may also serve as a potential biomarker to differentiate between the benign and malignant states of endometriosis and endometrial cancer.

## Figures and Tables

**Figure 1 biomolecules-14-00770-f001:**
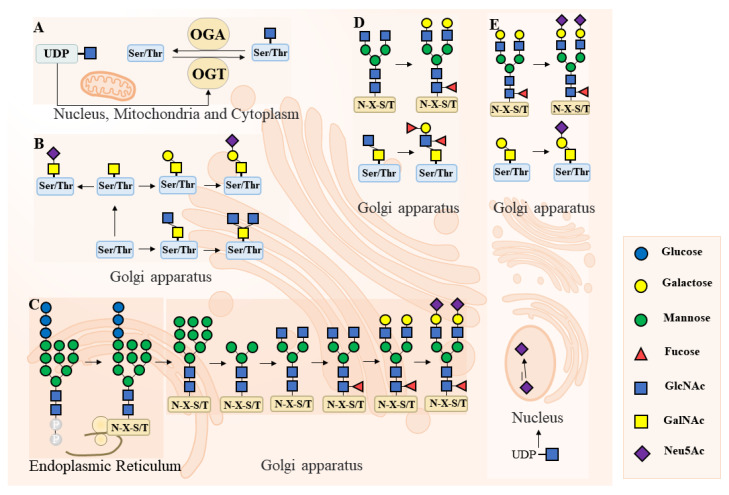
Protein glycosylation classification. Common protein glycosylation modifications in eukaryotic cells. (**A**) Nuclear, mitochondrial, and cytoplasmic proteins are modified by O-GlcNAc. (**B**) O-GalNAc glycosylation occurs primarily in the Golgi apparatus. (**C**) N-glycan biosynthesis occurs in the endoplasmic reticulum and the Golgi apparatus. (**D**) Fucosylation occurs in the Golgi apparatus and also modifies the ends of sugar chains. (**E**) Neu5Ac is biosynthesized in the cytoplasmic compartment of UDP-GlcNAc and converted into activated CMP-Neu5Ac in the nucleus, and in Golgi, Neu5Ac is added to the end of the sugar chain. Abbreviations for sugar involved: N-Acetyl-D-Glucosamine (GlcNAc), N-Acetyl-D-Galactosamine (GalNAc), and N-acetylneuraminic acid (Neu5Ac).

**Figure 2 biomolecules-14-00770-f002:**
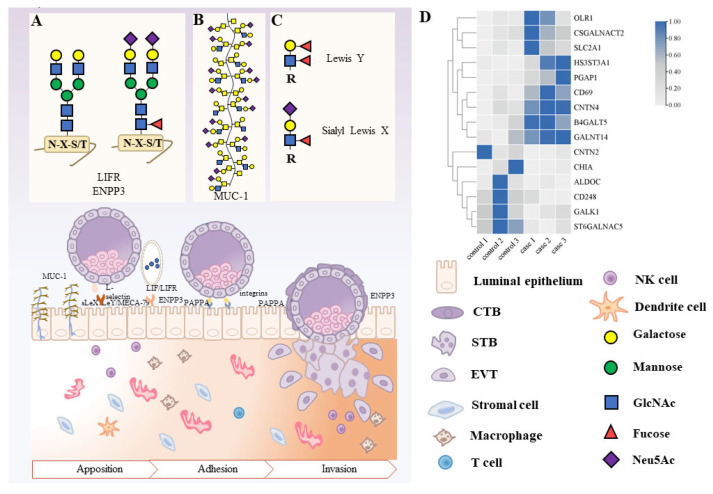
Protein glycosylation is involved in blastocyst recognition, adhesion, and invasion of the endometrium. (**A**) N-linked glycosylation of LIFR and ENPP3 in blastocyst localization and adhesion. (**B**) Mucins are a common O-linked glycosylation of proteins that play a role in blastocyst adhesion. (**C**) Lewis Y is regulated by fucosyltransferase, while sialyl Lewis X is sialylation modified, and both are involved in blastocyst implantation. (**D**) Expression of glycosylase in decidua tissue at 12 weeks. This is a comparison of decidual tissue mRNA expression profiles in seven RSA patients and seven healthy control women who underwent induced abortion. According to *p* < 0.05, the GEO database (GSE113790) for recurrent abortion and induced abortion found 15 differential genes related to glycosylation.

**Figure 3 biomolecules-14-00770-f003:**
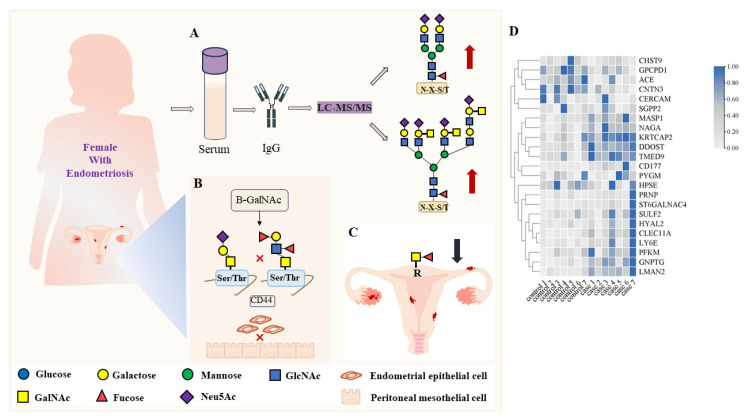
Current research progresses in glycosylation and endometriosis. (**A**) The double-antenna dichotomized monosialoglycan and high-branched chain N-glycan levels of IgG increased in the serum of females with endometriosis [79,80]. (**B**) The O-linked glycosylation of CD44 was inhibited, which significantly inhibited the adhesion of endometrial cells to peritoneal mesothelial cells [81]. (**C**) Fucosylated GalNAc decreased in women with advanced endometriosis [8]. (**D**) According to *p* < 0.05, the transcriptome analysis of the GEO database (GSE232713) of endometriosis and normal endometrium revealed 24 differential genes related to glycosylation. Compared with normal endometrium, most of the differential glucose genes in endometriosis are upregulated.

**Figure 4 biomolecules-14-00770-f004:**
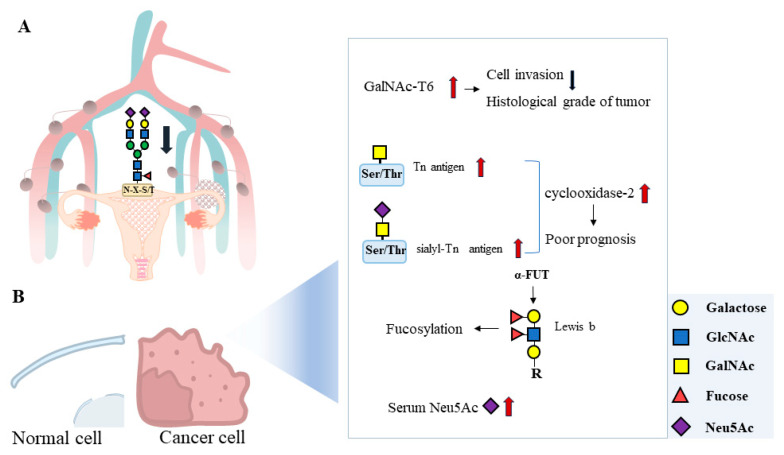
Protein glycosylation in the development of endometrial carcinoma. (**A**) In primary tumors with lymph node metastasis, the abundance of N-glycans in complex core concentrations is reduced. (**B**) Endometrial cancer cells have elevated levels of glycosylation compared to normal cells. Elevated GalNAc-T6 is associated with a lower invasive profile of endometrial cancer cells. Overexpression of Tn antigen and sialyl-Tn antigen increases the level of cyclooxidase-2, suggesting a poor prognosis. α-FUT regulates Lewis b levels, which in turn affects fucosylation in cancer cells. Serum Neu5Ac levels significantly increase in patients with endometrial cancer.

## Data Availability

No new data were generated or analyzed in support of this research.

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
