# Peer review of "The Sweet Relationship between the Endometrium and Protein Glycosylation"

_biomolecules, 2024, doi:10.3390/biom14070770_

Round 1

Reviewer 1 Report

Comments and Suggestions for Authors

The sweet relationship between endometrium and protein glycosylation

In this review, the authors highlight the role of protein glycosylation in embryo-maternal endometrial dialogue and explore the potential mechanisms of glycosylation in endometrial receptivity-related diseases. With an amazing title, the author made a good overview of the glycosylation and the endometrium. My overall recommendation is to accept the paper after the minor corrections.

Abstract

1)    “The incidence of endometrial cancer is increasing, and the glycosylation modification on the surface of tumor cells can be a marker to distinguish the type and severity of endometrial cancer.” I would write this phrase: In cancer, the glycosylation modification on the surface of tumor cells can be a marker to distinguish the type and severity of endometrial cancer.

Introduction

2)    Line 64: “dialogue between the embryo and endometrial.” Would not be endometrium?

3)    Line 68-69: “The occurrence and development of all these diseases are related to dysregulated glycosylation.” Please, add references for this phrase.

4)    Figure 1: Please, put Figure 1 close to where belongs. In this case, close to the line 87.

5)    Line 109: Please, add a final point and references in this phrase.

6)    Line 134: “As mentioned earlier, Sias is usually the most terminal sugar…” Maybe I missed it, but where is this mentioned?

Glycosylation and endometrial receptivity

7)    Line 139-145: This paragraph is repetitive. Please, rewrite it.

The relationship between endometriosis and glycosylation

8)    Line 310-311: “Endometriosis, the most common benign disease of endometrial, is the intrusion of 310 endometrial glands and stroma into the myometrium [76].” Sorry, but this is wrong. Endometriosis is characterized by the presence of epithelial, stromal, and muscle cells outside the uterine cavity. The definition that you gave is for adenomyosis. Please, correct it.

9)     Line 314: Please, add the reference at the end of this phrase: Mechsner S. Endometriosis, an Ongoing Pain-Step-by-Step Treatment. J Clin Med. 2022 Jan 17;11(2):467. doi: 10.3390/jcm11020467.

10)  Line 354: Human gene names should always be italicized as “ACE, HPSE, HYAL2, LMAN2 and SULF2”. Please, correct it.

11)  Line 358-360: What do “ACE 2350*G and ACE- 240* T” mean? If you are talking about a polymorphism, the correct way to write it is ACE c.2350A>G. Please, correct it.

Glycosylation of endometrial cancer cells

12)  Figure 4. The legend needs to be improved. Information is missing.

Conclusions

13)  Line 508-517: The message is good, but not how is written. Please, write it again. 

Comments on the Quality of English Language

Minor corrections are needed.

Author Response

Dear Reviewer

We thank you for the positive and constructive comments and suggestions. We have revised the manuscript according to the comments and responded point-by-point to the comments, as listed below. We highlighted the corresponding revisions in the manuscript.

My comments are the following:

In this review, the authors highlight the role of protein glycosylation in embryo-maternal endometrial dialogue and explore the potential mechanisms of glycosylation in endometrial receptivity-related diseases. With an amazing title, the author made a good overview of the glycosylation and the endometrium. My overall recommendation is to accept the paper after the minor corrections.

Abstract

1)    “The incidence of endometrial cancer is increasing, and the glycosylation modification on the surface of tumor cells can be a marker to distinguish the type and severity of endometrial cancer.” I would write this phrase: In cancer, the glycosylation modification on the surface of tumor cells can be a marker to distinguish the type and severity of endometrial cancer.

Response: Thanks for your suggestions. We have revised the sentence according to your suggestion.

As follows: “In cancer, glycosylation modification on the surface of tumor cells can be a marker to distinguish the type and severity of endometrial cancer.”

Introduction

2)    Line 64: “dialogue between the embryo and endometrial.” Would not be endometrium?

Response: Thank you for pointing out this error. We have changed ‘endometrial’ to ‘endometrium’.

As follows: “dialogue between the embryo and endometrium”.

3)    Line 68-69: “The occurrence and development of all these diseases are related to dysregulated glycosylation.” Please, add references for this phrase.

Response: Thanks for your suggestion, we added references.

As follows: “The occurrence and development of all these diseases are related to dysregulated glycosylation [5-12].”

4)    Figure 1: Please, put Figure 1 close to where belongs. In this case, close to the line 87.

Response: We've placed Figures where it belongs.

5)    Line 109: Please, add a final point and references in this phrase.

Response: Thanks for your suggestion, we added a final point and reference.

As follows: “Some sugar residues or their modifications may mask potential antigens or receptors [26].”

6)    Line 134: “As mentioned earlier, Sias is usually the most terminal sugar…” Maybe I missed it, but where is this mentioned?

Response: We mentioned this concept in lines 56-57.

As follows: “Because fucose and Sia, two terminal modifications, have been extensively studied in endometrial-related diseases, they are also discussed in separate sections.”

Glycosylation and endometrial receptivity

7)    Line 139-145: This paragraph is repetitive. Please, rewrite it.

Response: Thank you for pointing out this error. We've rewritten the paragraph.

As follows: “The study by Carson, D. D. (2002) showed that glycosylation can improve endometrial receptivity [33] (Figure 2). Endometrial receptivity refers to the ability of the endometrium to promote embryo attachment, implantation, and subsequent growth [34]. Acceptance of the endometrium is an important prerequisite for embryo implantation, which is a key step in pregnancy.”

The relationship between endometriosis and glycosylation

8)    Line 310-311: “Endometriosis, the most common benign disease of endometrial, is the intrusion of 310 endometrial glands and stroma into the myometrium [76].” Sorry, but this is wrong. Endometriosis is characterized by the presence of epithelial, stromal, and muscle cells outside the uterine cavity. The definition that you gave is for adenomyosis. Please, correct it.

Response: Thank you for pointing out this error. We've redefined endometriosis.

As follows: “Endometriosis is commonly thought to be caused by retrograde menstruation and is characterized by the presence of endometrium tissue (including epithelial, stromal, and muscle cells) outside the uterus [78]”

9)     Line 314: Please, add the reference at the end of this phrase: Mechsner S. Endometriosis, an Ongoing Pain-Step-by-Step Treatment. J Clin Med. 2022 Jan 17;11(2):467. doi: 10.3390/jcm11020467.

Response: Thanks for your suggestion, we added this reference.

As follows: “The early detection and diagnosis of endometriosis can help early treatment and reduce the symptoms of patients [80].”

10)  Line 354: Human gene names should always be italicized as “ACE, HPSE, HYAL2, LMAN2 and SULF2”. Please, correct it.

Response: Thank you for pointing out this error. We use italics for these gene names.

As follows: “Existing studies have indicated that ACE, HPSE, HYAL2, LMAN2, and SULF2 are related to endometriosis.”

11)  Line 358-360: What do “ACE 2350*G and ACE- 240* T” mean? If you are talking about a polymorphism, the correct way to write it is ACE c.2350A>G. Please, correct it.

Response: Thank you for pointing out this confusing naming. We amend “ACE 2350*G and ACE- 240* T” to “ACE c.2350A>G and ACE c.240A>T”.

As follows: “The polymorphism of ACE c.2350A>G and ACE c.240A>T genes may be associated with the development of endometriosis [88].”

Glycosylation of endometrial cancer cells

12)  Figure 4. The legend needs to be improved. Information is missing.

Response: Thanks for your suggestion, we have improved the legend of Figure 4.

As follows: “Figure 4. Protein glycosylation in the development of endometrial carcinoma. (A) In primary tumors with lymph node metastasis, the abundance of N-glycans in complex core concentrations is reduced. (B) Endometrial cancer cells have elevated levels of glycosylation compared to normal cells. Elevated GalNAc-T6 is associated with a lower invasive profile of endometrial cancer cells. Overexpression of Tn antigen and sialyl-Tn antigen increases the level of cyclooxidase-2, suggesting a poor prognosis. α-FUT regulates Lewis b levels, which in turn affects fucosylation in cancer cells. Serum Neu5Ac levels significantly increase in patients with endometrial cancer.”

Conclusions

13)  Line 508-517: The message is good, but not how is written. Please, write it again.

Response: Thanks for your suggestion, we have improved this message.

As follows: “In conclusion, the current research on the correlation between glycosylation and endometrial receptivity provides valuable insights for the clinical diagnosis and management of spontaneous miscarriage. Furthermore, anomalies in glycosylation targets could form the basis for the development of innovative contraceptive approaches. Glycosylation intervention holds promise for the amelioration of ectopic colonization of endometrial cells and the mitigation of endometrial cancer progression. Notably, glycosylation may also serve as a potential biomarker to differentiate between the benign and malignant states of endometriosis and endometrial cancer.”

Reviewer 2 Report

Comments and Suggestions for Authors

Line 88: N-glycan is covalently attached to proteins on asparagine (Asn) residues via “N-glucoside bonds”. Should be “N-glycosidic bond”.

Line 114: It is catalyzed by “L-focus kinase” and “GDP-L-focus” pyrophosphorylase to convert “focus to GDP-focus” through a two-step mechanism. Replace “focus” by fucose. Please check all along the text.

Line 120: “These FUTs are localized to the Golgi apparatus and are fucosylated to oligosaccharides by α (1,2) -linkage with terminal galactose or α(1,3/4) -linkage with subterminal GlcNAc to produce Lewis antigens [22].” This sentence is difficult to read.

Line 139: “Recent study has shown that glycosylation can improve endometrial receptivity, thereby promoting embryo localization, adhesion, and invasion, ultimately contributing to successful implantation [31].” The reference is from 2002, 22 years ago, not really “recent”.

Line 198: “GLUT1 can promote the acylation of O-GlcNAc via HBP, which is the most important source of O-GlcNAc [51].” GLUT1 is a transporter of glucose, thus, it does not promote any acylation reaction. It should also be clarified what the authors mean with acylation of O-GlcNAc. How the GlcNAc is aceylated?

Line 204: “In addition, OGA inhibitors can increase the acylation of O-GlcNAc by reducing the amount of O-GlcNAc being hydrolyzed by proteins [53].” Again, please revise this sentence.

Line 259: “Recent studies have shown that the effect of Sialylation on endometrial receptivity mainly involves the interaction between sLeX and selection.” Reference is required. What is selection? Maybe Selectin?

Line 261: “sia” should be “Sia” and the authors should be sure that they have previously defined “Sia” as abbreviation of sialic acid.

Line 262: “Selectin (L-selectin, e-selectin, and p-selectin) [66] is the main ligand of sLeX and is involved in fertilization [67] and implantation [68].” Please restyle this sentence. sLeX is the ligand of Selectin, not the other way around.

Line 321: “Besides, compared with healthy women, IgG isolated from the advanced endometriosis group showed significantly higher expression of high-branched-N-glycan-reactive multi-antenna N-glycan.” Please explain what are “high-branched-N-glycan-reactive multi-antenna N-glycan”, why “reactive”?

Line 330: “B-GalNAc inhibits the O-linked glycosylation of CD44”. What is B-GalNAc? By reading the related article, reference [80], I learned that B-GalNAc is benzyl 2-acetamido-2-deoxy-α-D-galactopyranoside. Thus, the next question is How a synthetic molecules such as B-GalNAc “plays a role in the establishment of early endometriosis lesions”(line 332)? I mean, in the articles the authors used B-GalNAc and tunicamycin to inhibit CD44 O- and N-glycosylation, respectively. In such a way, they demonstrated that CD44 glycosylation is essential for endometrial cell attachment to peritoneal mesothelial cells. This is different from what the authors of this review stated.

Line 366: “Expression of heparinase-1, an endo-glycosidase that degrades heparin sulfate proteoglycan, in endometrial tissues of patients with endometriosis.” There is not a verb in this sentence. Please revise.

Line 412: “Tunicamycin reduced the concentration of MUC1 on the cell surface more than benzyl glycoside and greatly reduced glycosylation of the glycoprotein, resulting in increased cell adhesion of cancer cells [14].” Tunicamycin is an inhibitor of protein N-glycosylation, but not of protein O-glycosylation. I do not see how Tunicamycin can reduce the concentration of MUC1 since MUC1 is a o-glycoprotein and not N-. Please clarify.

Line 447: “focusing (core-focusing)”. Please revise “focusing”.

Line 470: “Sialylation occurs primarily in the nucleus, where the Golgi is added to the end of the sugar chain.” This sentence does not make sense.

Figure 1: Please indicates what does means the UDP with sialic acid and the Nucleus in the lower-right part of the figure.

Line 502: This may be due to the complexity of the glycose structure [1]”. What is glycose structure??

Please revise figure legends. In figure 3, (A) panel is defined as “N-linked glycosylation levels of IgG in the serum of female with endometriosis”. However, it only reports two N-glycan structures. Similarly, panel (B) defined as “B-GalNAc inhibits the adhesion of endometrial cells to peritoneal mesothelial cells by inhibiting O-linked glycosylation of CD44”, yet, it only show the cartoon structure of a MUC1 protein. Panel (C) as “Fucosylated GalNAc decreased in women with advanced endometriosis” however it reports two glycan structures and none of them presents the fucose linked to GalNAc.

Please revise also the language of the figure’s footnote and ensure that all panels are discussed and indicated in the figure.

Comments on the Quality of English Language

Quality of English language should be improved. I indicated in the main comments file some of the sentences that are really difficult to read.

Author Response

Dear Reviewer

Thank you for taking the time to review our manuscript. First of all, we would like to express our sincere gratitude to you for the constructive and insightful comments. These suggestions have greatly improved the manuscript. We revised the manuscript based on the comments and responded to each comment individually, as shown below. In addition, we have highlighted the text in the revised manuscript.

Line 88: N-glycan is covalently attached to proteins on asparagine (Asn) residues via “N-glucoside bonds”. Should be “N-glycosidic bond”.

Response: Thank you for pointing out this error. We have changed “N-glucoside bonds” to“N-glycosidic bond”.

As follows: “N-glycan is covalently attached to proteins on asparagine (Asn) residues via N-glycosidic bonds, which is called the N-linked glycosylation modification of proteins.”

Line 114: It is catalyzed by “L-focus kinase” and “GDP-L-focus” pyrophosphorylase to convert “focus to GDP-focus” through a two-step mechanism. Replace “focus” by fucose. Please check all along the text.

Response: Thank you. We apologize for not finding these errors. We have checked all along the text.

Line 120: “These FUTs are localized to the Golgi apparatus and are fucosylated to oligosaccharides by α (1,2) -linkage with terminal galactose or α(1,3/4) -linkage with subterminal GlcNAc to produce Lewis antigens [22].” This sentence is difficult to read.

Response: Thanks for your suggestion. We recalibrated and streamlined the sentence structure.

As follows: “These FUTs are localized to the Golgi apparatus. Lewis antigens are produced by adding fucose to the terminal galactose and subterminal GlcNAc of oligosaccharides.”

Line 139: “Recent study has shown that glycosylation can improve endometrial receptivity, thereby promoting embryo localization, adhesion, and invasion, ultimately contributing to successful implantation [31].” The reference is from 2002, 22 years ago, not really “recent”.

Response: Thank you for pointing out this error. We have changed “Recent study” to “A study”.

As follows: “The study by Carson, D. D. (2002) showed that glycosylation can improve endometrial receptivity.”

Line 198: “GLUT1 can promote the acylation of O-GlcNAc via HBP, which is the most important source of O-GlcNAc [51].” GLUT1 is a transporter of glucose, thus, it does not promote any acylation reaction. It should also be clarified what the authors mean with acylation of O-GlcNAc. How the GlcNAc is aceylated?

Response: Thank you for pointing out this error. Elevated O-GlcNAcylation increases glucose uptake through glucose transporter 1 (GLUT1), leading to glucose metabolic flow into the hexosamine biosynthesis pathway (HBP), thereby regulating metabolic reprogramming of endometrial cells. Activation of endometrial HBP leads to increased O-GlcNAcylation during the implantation window. Therefore, GLUT1 does not directly affect O-GlcNAcylation, but rather an indirect effect. Our description is inaccurate, so we have amended the corresponding section.

As follows: “GLUT1 can increase glucose uptake by endometrial cells into the hexosamine biosyn-thesis pathway (HBP), and the activation of HBP increases O-GlcNAcylation.”

Line 204: “In addition, OGA inhibitors can increase the acylation of O-GlcNAc by reducing the amount of O-GlcNAc being hydrolyzed by proteins [53].” Again, please revise this sentence.

Response: Thanks for your suggestion. We have revised this sentence.

As follows: “In addition, OGA inhibitors can inhibit the hydrolysis of O-GlcNAc, thereby increasing the level of O-GlcNAc.”

Line 259: “Recent studies have shown that the effect of Sialylation on endometrial receptivity mainly involves the interaction between sLeX and selection.” Reference is required. What is selection? Maybe Selectin?

Response: Thanks for your suggestion. We have added references and changed “selection” to“selectin”.

As follows: “Recent studies have shown that the effect of sialylation on endometrial receptivity mainly involves the interaction between sLeX and selectin [68-70].”

Line 261: “sia” should be “Sia” and the authors should be sure that they have previously defined “Sia” as abbreviation of sialic acid.

Response: Thank you for pointing out this error. We have defined “Sia” as an abbreviation of sialic acid where it first appears. In addition, we checked the entire manuscript and revised it accordingly.

Line 262: “Selectin (L-selectin, e-selectin, and p-selectin) [66] is the main ligand of sLeX and is involved in fertilization [67] and implantation [68].” Please restyle this sentence. sLeX is the ligand of Selectin, not the other way around.

Response: Thank you for pointing out this error. We have revised this sentence.

As follows: “sLeX is the main ligand of selectin (L-selectin, E-selectin, and P-selectin) [71] and is involved in fertilization [72] and implantation [68].”

Line 321: “Besides, compared with healthy women, IgG isolated from the advanced endometriosis group showed significantly higher expression of high-branched-N-glycan-reactive multi-antenna N-glycan.” Please explain what are “high-branched-N-glycan-reactive multi-antenna N-glycan”, why “reactive”?

Response: Thank you for pointing out this error. We have revised this conjunction. The “high-branched-N-glycan-reactive multi-antenna N-glycan” should be “PHA-L-reactive multi-antennary N-glycans”. PHA-L is a lectin that recognizes highly branched N-glycans.

As follows: “Additionally, compared to healthy women, IgG isolated from the advanced endome-triosis group showed a significantly higher expression of highly branched multi-antennary N-glycans.”

Line 330: “B-GalNAc inhibits the O-linked glycosylation of CD44”. What is B-GalNAc? By reading the related article, reference [80], I learned that B-GalNAc is benzyl 2-acetamido-2-deoxy-α-D-galactopyranoside. Thus, the next question is How a synthetic molecules such as B-GalNAc “plays a role in the establishment of early endometriosis lesions”(line 332)? I mean, in the articles the authors used B-GalNAc and tunicamycin to inhibit CD44 O- and N-glycosylation, respectively. In such a way, they demonstrated that CD44 glycosylation is essential for endometrial cell attachment to peritoneal mesothelial cells. This is different from what the authors of this review stated.

Response: Thank you for pointing out this error. We apologize for not noticing the problem, and after a careful reading of the article, we found that it should be O-linked glycosylation of CD44 plays a role in the establishment of early endometriosis lesions, not B-GalNAc. B-GalNAc is benzyl 2-acetamido-2-deoxy-α-D-galactopyranosid, an O-linked glycosylation inhibitor.

As follows: “B-GalNAc inhibits the O-linked glycosylation of CD44, thus significantly inhibiting the adhesion of endometrial cells to peritoneal mesothelial cells, suggesting that the O-linked glycosylation of CD44 plays a role in the establishment of early endometriosis lesions [83].”

Line 366: “Expression of heparinase-1, an endo-glycosidase that degrades heparin sulfate proteoglycan, in endometrial tissues of patients with endometriosis.” There is not a verb in this sentence. Please revise.

Response: Thank you for pointing out this error. We have revised this sentence.

As follows: “Heparinase-1 (an endo-glycosidase that degrades heparin sulfate proteoglycans) is expressed in the endometrial tissue of patients with endometriosis.”

Line 412: “Tunicamycin reduced the concentration of MUC1 on the cell surface more than benzyl glycoside and greatly reduced glycosylation of the glycoprotein, resulting in increased cell adhesion of cancer cells [14].” Tunicamycin is an inhibitor of protein N-glycosylation, but not of protein O-glycosylation. I do not see how Tunicamycin can reduce the concentration of MUC1 since MUC1 is a o-glycoprotein and not N-. Please clarify.

Response: Thank you for pointing out this error. This sentence is a reference from the article (Paszkiewicz-Gadek, A.; Porowska, H.; Lemancewicz, D.; Wolczynski, S.; Gindzienski, A., The influence of N- and O-glycosylation inhibitors on the glycosylation profile of cellular membrane proteins and adhesive properties of carcinoma cell lines. Int J Mol Med 2006, 17, (4), 669-74.). However, tunicamycin does act as an inhibitor of N-linked glycosylation, and MUC1 is a common O-glycoprotein, so we also think this statement is wrong. We are sorry that we did not notice it at the time and have now removed this sentence and the related reference.

Line 447: “focusing (core-focusing)”. Please revise “focusing”.

Response: Thank you for pointing out this error. We have changed “focusing” to “fucosing”.

Line 470: “Sialylation occurs primarily in the nucleus, where the Golgi is added to the end of the sugar chain.” This sentence does not make sense.

Response: Thank you for your advice. We have revised this sentence.

As follows: “Neu5Ac is biosynthesized in the cytoplasmic compartment of UDP-GlcNAc and converted into activated CMP-Neu5Ac in the nucleus, and in Golgi, Neu5Ac is added to the end of the sugar chain.”

Figure 1: Please indicates what does means the UDP with sialic acid and the Nucleus in the lower-right part of the figure.

Response: Thank you for pointing out the ambiguity. Neu5Ac is biosynthesized in the cytoplasmic compartment of UDP-GlcNAc. I have added this explanation to the notes and modified the corresponding places in the Figure 1.

As follows: “Neu5Ac is biosynthesized in the cytoplasmic compartment of UDP-GlcNAc and converted into activated CMP-Neu5Ac in the nucleus, and in Golgi, Neu5Ac is added to the end of the sugar chain.”

Line 502: This may be due to the complexity of the glycose structure [1]”. What is glycose structure??

Response: Thank you for pointing out this error. This is supposed to be the “structure of the sugar chain”, and we modified it.

As follows: “This may be due to the complexity of the structure of sugar chains.”

Please revise figure legends. In figure 3, (A) panel is defined as “N-linked glycosylation levels of IgG in the serum of female with endometriosis”. However, it only reports two N-glycan structures. Similarly, panel (B) defined as “B-GalNAc inhibits the adhesion of endometrial cells to peritoneal mesothelial cells by inhibiting O-linked glycosylation of CD44”, yet, it only show the cartoon structure of a MUC1 protein. Panel (C) as “Fucosylated GalNAc decreased in women with advanced endometriosis” however it reports two glycan structures and none of them presents the fucose linked to GalNAc. Please revise also the language of the figure’s footnote and ensure that all panels are discussed and indicated in the figure.

Response: Thank you for pointing out these mistakes. We have modified the corresponding part.

In Panel (A), we modified N-linked glycosylation to two specific N-glycans. Panel (B), the area of the figure is limited, and we only show some of the common forms of O-linked glycosylation. (C) Reference mentioned that DBA lectin can identify fucosylated N-acetylgalactosamine (GalNAc) sequences. As shown in the Panel (C). Besides, we revised the description of the figure’s footnote and ensured that all panels are discussed and indicated in the figure.

As follows: “Figure 3. Current research progress on glycosylation and endometriosis. (A) The double-antenna dichotomized monosialoglycan and high-branched chain N-glycan levels of IgG increased in the serum of female with endometriosis [81, 82]. (B) The O-linked glycosylation of CD44 was inhibited, which significantly inhibited the adhesion of endometrial cells to peritoneal mesothelial cells [83]. (C) Fucosylated GalNAc decreased in women with advanced endometriosis [8].”

Quality of English language should be improved. I indicated in the main comments file some of the sentences that are really difficult to read.

Response: We have asked a professional team to polish the language. See proof of language polishing as follows.

Round 2

Reviewer 2 Report

Comments and Suggestions for Authors

The review manuscript improved significantly. It well describe the current status of the field and is an important contribution.